# A Baseline Study of Emphasis Effects in Information Visualization

Aristides Mairena *   Martin Dechant †   Carl Gutwin ‡   Andy Cockburn §

University of Saskatchewan
University of Canterbury

## ABSTRACT

Emphasis effects – visual changes that make certain elements more prominent – are commonly used in information visualization to draw the user's attention or to indicate importance. Although theoretical frameworks of emphasis exist (that link visually diverse emphasis effects through the idea of visual prominence compared to background elements), most metrics for predicting how emphasis effects will be perceived by users come from abstract models of human vision which may not apply to visualization design. In particular, it is difficult for designers to know, when designing a visualization, how different emphasis effects will compare and how to ensure that the user's experience with one effect will be similar to that with another. To address this gap, we carried out two studies that provide empirical evidence about how users perceive different emphasis effects, using three visual variables (colour, size, and blur/focus) and eight strength levels. Results from gaze tracking, mouse clicks, and subjective responses in our first study show that there are significant differences between different kinds of effects and between levels. Our second study tested the effects in realistic visualizations taken from the MASSVIS dataset, and saw similar results. We developed a simple predictive model from the data in our first study, and used it to predict the results in the second; the model was accurate, with high correlations between predictions and real values. Our studies and empirical models provide new information for designers who want to understand how emphasis effects will be perceived by users.

**Index Terms:** Human-centered computing—Visualization—Visualization techniques—Perception; Human-centered computing—Visualization—Visualization design and evaluation methods

## 1 INTRODUCTION

Emphasis effects are visual changes that make certain elements more prominent, and are commonly used in information visualization to draw the user's attention or to indicate importance. Emphasizing important data points is a common method used by designers to support the user when gradually exploring the data – or in narrative visualization [25], when known aspects of the data are presented to the users. Effective emphasis alters a data point's visual features [4, 22] so that a viewer's attention will be guided to the region of interest [62]. A wide variety of visual variables have been considered as emphasis effects [17, 22, 23, 60]. For example, a visualization can use colour to emphasize certain data points: differences in the visual prominence of the selected data points will be achieved through the variation in color, a visual variable known to guide attention [24].

Although theoretical frameworks of emphasis exist that link visually diverse emphasis effects through the idea of visual prominence compared to background elements [22], we still know little about

*e-mail: aristides.mairena@usask.ca

†e-mail:martin.dechant@usask.ca

‡e-mail:gutwin@cs.usask.ca

§e-mail:andy@cosc.canterbury.ac.nz

Graphics Interface Conference 2020
28-29 May

how emphasis effects will be perceived by users. In particular, questions remain about what visual effects, and what magnitudes of those effects, will be most quickly recognized as emphasis by the viewer of a visualization; in addition, we know little about how different effects compare and what levels of two different effects will lead to similar user experiences.

Many metrics for predicting how emphasis effects will be perceived by users come from abstract models of human vision which may not apply to visualization design. These models are generally constructed using large and visually isolated stimuli under optimal conditions, and so although they are effective at predicting perceptibility in isolation, they do not work well with distractors, and do not work well with even minor changes to the visual field [3, 59].

Visualizations, in contrast, often consist of large numbers of a variety of marks viewed using a wide range of devices and environments — and designers may use a variety of techniques to emphasize data points. Current guidelines do not address how different emphasis effects are perceived by viewers in visualizations, or provide an equivalence metric for perceived emphasis so designers can choose effects correctly. Without effective models of visual prominence in visualizations, designers lack information on how different visual effects compare, and do not know what magnitude of effect to use to appropriately guide a viewer's attention to an area of interest.

To address this gap, we carried out two studies that provide empirical evidence about how users perceive different emphasis effects, using three visual variables (colour, size, and blur/focus) and eight strength levels. It is important to note that the three emphasis effects are qualitatively different — for example, colour and size manipulate just the emphasized element, whereas blur/focus manipulates everything but the emphasized element - and so our goal is not simply to identify which effect is most perceivable, but rather to establish how the effects compare across a range of intensity levels.

To do this, our first study established a baseline of perceived visual prominence by showing participants artificial scatterplot visualizations with one element highlighted, and measuring the time to their first fixation on the element (through eye tracking), their time to click on the element, and their subjective rating of visual prominence. We then built a model from the first study's data using logarithmic curves, that can be used to predict the relationship between the different emphasis effects. Our second study then examined perceived emphasis in a more realistic context, by looking at visual prominence in complex visualizations that are taken from real-world applications (the MASSVIS dataset [7]). We evaluated our model by using it to predict the results of the second study for three different measures; the model was accurate, with $R^2$ values as high as 0.96.

Our two studies provide new findings about how people perceive three emphasis effects and their magnitudes in visualizations:

- There were significant differences in both studies for emphasis effect: blur/focus was most prominent, and colour least prominent, with size in between depending on magnitude.

- There were also significant differences between the magnitude levels for all effects, providing a graduated way to increase or decrease perceived prominence.

- A predictive model based on logarithmic curves fit the Study 1 data well, and was accurate at predicting perceived emphasis in Study 2 (particularly in terms of subjective ratings).

It is important to note that our goal in the study was not to conduct a "shoot-out" between all possible types of emphasis, but rather to determine whether different effects are perceived differently, and to provide initial empirical evidence about how effects compare. Our results provide an initial empirical foundation for understanding how visual effects operate and are experienced by viewers when used for emphasis in visualizations; and although more work is needed to refine and broaden the models, our work provides useful new information for designers who want to control how emphasis effects will be perceived by users.

## 2 RELATED WORK

Emphasis is essential to InfoVis and is used to highlight regions of interest in a visualization. While there is a large body of research in this domain, much of the work seeks to understand how the underlying perceptual system operates – limiting the possibility of extracting design lessons from low-level data and findings. We survey current empirical studies of perception from visualization and vision science to inform our work.

### 2.1 Visual Attention and Graphical Perception

There are various theories and computational models for selective visual attention, but in general, most theories agree that attention operates by alternately selecting "features" from a number of incoming subsets of sensory data for further processing [51]. Early work suggests a two-stage process: first, a bottom-up, pre-attentive stage, which is automatic and independent of a task [51], where attention is guided to the most salient items in a scene [70]; followed by a second, slower, top-down stage that is driven by current tasks and goals [34, 50, 62]. Within this model, the conjunction of basic features (such as colour and orientation) stems from "binding" features together (known as Feature Integration Theory [62]).

A second theory, the Guided Search Theory, extends the two-stage process by proposing that attention can be biased toward targets of interest (e.g., a user looking for a red circle) in the top-down phase by encoding particular visual characteristics [69]: for example, assigning a higher weight to the red colour. Recently-proposed attention theories, however, challenge the two-stage model suggesting there is also bias to prioritize items that have been previously selected, thus proposing a three-stage model: current goals, selection history, and physical salience (bottom-up attention) [1].

Attention is commonly examined through visual search experiments which usually ask participants to determine whether a target is present in a scene with distractors; reaction times (RT) and accuracy are used to model the relationship between the response and the number of distractors. Frequently, the term "popout" is used to describe a target item that is easily identified due to its unique visual properties in these searches. While the noticeability of specific visual characteristics such as colour and size cues have also been examined from an attention perspective [21, 44], a related area of research called *graphical perception* takes a more in-depth look at the suitability of different visual channels, and at how choices in visual variables for encoding data affect visualization effectiveness [15].

Graphical perception studies have explored how different visual channels might support a variety of tasks for visualization [52]. Bertin was among the first to study the ability of visual variables to encode information, suggesting that variations in individual visual variables is an effective tool for encoding information and achieving noticeability [4]. Particularly, Bertin suggests that selective visual variables, such as position, size, colour hue, or texture allows viewers to immediately detect variables.

Following Bertin, researchers in multiple disciplines such as cartography [42], statistics [15], and computer science [43] have conducted human-subjects experiments and have derived rankings of visual variables for nominal, ordinal, or quantitative data [15, 42, 43, 57]. In addition to comparing the effectiveness of alternative visual variables for visualization, researchers have investigated how other design factors such as aspect ratios [13], chart sizes [26], and animations [63] influence the effectiveness of charts.

Graphical perception studies have focused on measuring how the visual encoding of variables affect the accuracy of estimating and understanding values of the underlying data; insights from studies in graphical perception, however, can also be applied in manipulating data points in a visualization to guide a viewer's attention to an area of particular importance.

### 2.2 Emphasis in Visualization

Emphasis is essential to information visualization by offering support to a user when exploring data, for instance through highlighting areas of interest when brushing and linking across multiple views to emphasize relationships [35]. Emphasis is also important when presenting known aspects of data to a user through narrative visualization [54]. The goal of emphasis is to manipulate the visual features of an important data point to make it visually prominent, such that a viewer's bottom-up attention is attracted to the point [22].

While distortion and magnification techniques – which create emphasis effects by simultaneously manipulating a visual variable's size and positioning – have been a focus of infovis researchers for creating emphasis [11, 32, 39], other techniques such as blur [36], motion [27], and flicker [65] have also been studied. Given the varied range of emphasis techniques, Hall *et al.* suggested a categorization of emphasis effects into two main groups based on how visuals change over time: time-invariant and time-variant effects [22].

Time-invariant emphasis effects such as highlighting (colouring a data point in a visualization), and blurring (where a data point is shown in focus while the other elements are blurred) do not change with time, and do not use features such as fly-in, fade-in or other transitions [22]. In contrast, time-variant emphasis effects such as motion, flickering, or zooming involve time variations, commonly achieved through animations that alter the appearance of a data point [22].

While there are many ways in which a data point in a visualization can be emphasized, all visual techniques generate emphasis by making the focus mark (i.e., the target) visually more prominent by making it sufficiently dissimilar from the other elements (i.e., the non-target marks) in at least one visual channel [22]. For example, blur/focus, magnification, and highlighting create emphasis by making one data point more visually prominent than others (e.g., sharper, bigger, or a different colour).

There are three main properties of a visual channel that could influence the effectiveness of the visual prominence of an emphasized target mark against the set of non-emphasized marks: the similarity between targets and non-targets, the similarity of all non-targets, and the channel offset (i.e., the lowest value of the non-targets) [64]. Similarity theory shows that visual search efficiency decreases with increased target/non-target similarity and with decreased similarity between the non-targets [18]).

Another theory, the relational account of attention theory, also suggests that the perceived similarity between targets and non-targets can be modeled by the magnitude of a vector in feature space pointing from the target to the closest non-target [2]. If users are given a feature direction in a visual search task (e.g., find the brightest or largest), attention will be guided to the mark that differs in the given direction from the other marks. In this theory, however, the non-target similarity does not have an influence on the visual prominence of a target.

Findings from classic psychophysics and visual search experiments, however, cannot always be applied directly to data visualization. Simple changes such as adding links between dots to simulate

a node-link diagram, or changes to contrast effects due to a background luminance have been shown to have considerable effects on the results from prior experiments [3,59]. These results reinforce the need for empirical evaluations of visualizations to validate theory and evaluate real-world visualization applications.

## 2.3 Evaluations of Perception in Visualization

Evaluations of perception in visualization have focused on understanding the details of integral and separable channels [58] and the interactions between separable channels. Smart and Szafir found that separability among shape, colour, and size perception functions asymmetrically, with shape found to affect the perception of size and colour more strongly compared to size's or colour's effect on shape perception [58]. Other studies have shown that size perception is biased by specific hues, and quantity estimation in visualizations are affected by both size and colour [14, 16].

Understanding of visual channels in visualization has largely categorized visual channels with terms such as "fully separable" and "no significant interference" [45, 70]. Design of visualizations that utilize these separable visual channels can be achieved by encoding data with visual variables that are known to "pop out". In a review of visual popout, Healey identified sixteen different visual variables or features that are known to pop out [24], including hue, size, orientation, and luminance.

Scatterplots are one of the most effective visualizations for visual judgments due to data points being positioned along a common scale [25]. Several studies have explicitly explored graphical perception in scatterplots, with many recent techniques being developed to automate scatterplot design [12], and to predict perceptual attributes that may affect scatterplot analysis such as similarity or separability. However, these studies and techniques primarily focus on analyses over single-channel features for scatterplot design to improve legibility or its suitability for data comparison [19].

Eye-tracking evaluations are a popular and effective tool for understanding how users view and visually explore visualizations [5,6]. For example, eye-tracking has been used to understand how different tasks and visual search strategies affect cognitive processes through fixation patterns [46,47], and has also been used to evaluate specific visualization types [10,29,30], for comparing multiple types of visualizations [20], and for evaluating decision making and interaction in visualization [6,33].

Free-viewing is a common technique for evaluating human perception of visual stimuli. Participants are not given a task and are instructed to freely look around the image, which avoids task-dependent effects and peripheral-vision effects. As some attention theories suggest that attention can be guided by a high-level task [1,70], free-viewing allows attention to be guided by image elements in a bottom-up manner. This assumption has guided researchers to the use of free-viewing for collecting ground truth data for evaluating saliency and attention in visualizations.

However, despite the extensive body of research from vision science on graphical perception, prior research has been focused on evaluating factors that may affect the visual prominence of a specific emphasis effect [64], or in empirically ranking visual variables for encoding data [15,26]; few guidelines discuss the issue of how different emphasis effects are perceived by viewers in visualizations, or consider issues of equivalence for perceived emphasis. Therefore, in the evaluations described next, we set out to determine the viewer's perception of visual prominence, and the effectiveness of a variety of emphasis effects at a wide range of intensity levels.

## 3 GENERAL STUDY METHODS

Data visualizations are used both to reveal patterns in data through exploration, and to communicate specific information to a viewer. When building visualizations for communication, a designer may need to draw a user's attention to a specific data point in order to better reveal the narrative focus of the visualization, and this can effectively be done by increasing the perceptual difference in the visual variables of the underlying data.

In the following two studies, we experimentally evaluate how specific emphasis effects are experienced by a viewer. Our first study was designed to determine the baseline visual prominence of eight levels of three emphasis effects using different visual variables (blur/focus, colour, and size). In simple scatterplot visualizations, we visually emphasized one data element, and gathered eye-tracking data, mouse clicks, and subjective ratings of visual prominence. Our second study built on the first; it used a similar paradigm but increased the complexity of the visualizations by using subset of the MASSVIS dataset – a repository of static data visualizations obtained from a variety of publicly-available online sources intended for a wide audience [7]. In addition, we developed predictive models from the first study's data, and used these to predict the results of the second study.

Our theoretical starting point for these studies was the mathematical framework of emphasis effects in data visualizations developed by Hall *et al.*, where visually diverse emphasis effects can be linked through the idea of visual prominence compared to background elements [22]. Our first study extends this previous work to determine the visual prominence of emphasis effects through eye-tracking metrics, click data, and subjective ratings. Using eye movement data makes it possible to examine which areas of a visualization viewers attend to and how their attention can be guided by applying emphasis effects. Combining eye tracking, interaction logs and subjective methods allowed us to collect a more diverse set of data, allowing us to analyze how participants' actions were guided by their perception of the different effects. This rich data allows us to better understand how users perceive commonly used effects that designers can use to emphasize a particular element in a visualization.

## 3.1 Emphasis Effects and Levels

Many modern visualization software and libraries utilize a wide range of emphasis techniques. For example, Chart.js, a commonly-used visualization library for the web, increases a mark's size when a user clicks on it to generate emphasis, while Tableau uses a combination of blur/focus and size to emphasize a mark. Based on an informal survey of visualization tools, we chose three visual variables for our study – colour, blur/focus, and size – that are commonly used to provide emphasis in many different contexts. While other common techniques exist (such as border highlighting), the variables we chose are known to be detected and processed in a pre-attentive way [24].

- *Colour.* Emphasizing an element using colour means changing the hue of the data element to be different from the standard element colour; colour is well known to "pop out" when there is adequate difference between the highlighted item and the other elements, and colour change is widely used to indicate importance.

- *Size.* Emphasizing an element using size means increasing the area of the data element such that it is bigger than other elements. Size also pops out, and is used in several visualization tools for interactive highlighting.

- *Blur/Focus.* Emphasizing an element using blur/focus means applying a blur filter (e.g., Gaussian blur) to all of the elements in the visualization except for the emphasized element (which remains sharp). This effect is therefore qualitatively different from colour and size because it affects a much larger fraction of the overall view.

For each type of emphasis effect, we chose several levels of the visual variable so that we could test the effect at different levels of

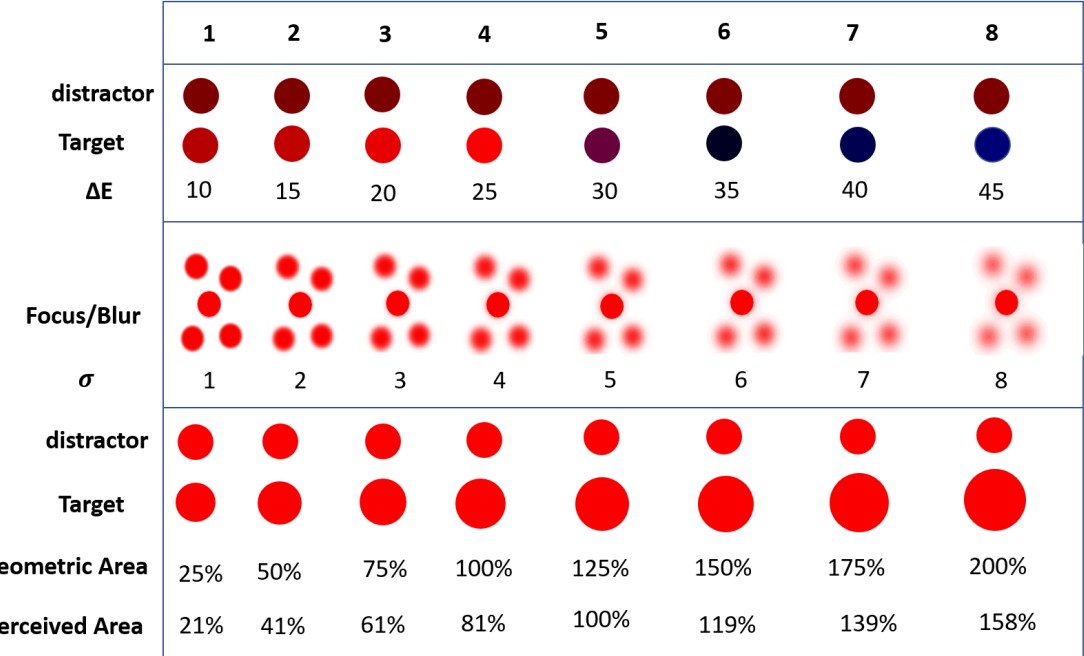

Figure 1: Visual variables and magnitude of difference levels. Rows 1-3: Colour, Blur/Focus, Size (distractors upper, targets lower).

magnitude (eight levels for Study 1, and three levels for Study 2). We sampled mark sizes, colour differences, and blur strength along increasing levels of difference between the target and the distractor – we call these levels 'magnitude of difference'. For some of the visual variables, the magnitude of difference range was constrained at both ends (e.g., there is a fixed range of hues between red and blue); for other variables, such as blur or size, the range was constrained only at one end (e.g., blur/focus and size start from the sharpness and size of the distractors and range up to an arbitrary upper end).

It is important to note that the magnitude scales for each variable are different, since we do not have a way to translate perceptual equivalency across effects (investigating this equivalency is one of the goals of our studies). For colour, we chose eight magnitude levels using a colour difference metric that normalizes the colour space to provide a closer fit between perceptual and geometric differences between colours [49]. $\Delta E$ is a metric devised to understand and measure how the human eye perceives colour difference, where a difference of 2.3 is roughly equal to one Just Noticeable Difference (JND) [55]. By utilizing $\Delta E$, we can more accurately compare a wider range of colours, utilizing all the colours of a colour space to compare differences and comparing their change in visual perception. We use the current $\Delta E$ standard, CIEDE2000 [56], as our primary colour difference metric, which has added corrections to account for lightness, chroma, and hue. For the colour levels used in the first study, we chose eight fixed colour differences (i.e., the difference between emphasized and non-emphasized elements) ranging from $\Delta E$ 10 to $\Delta E$ 45 (see Figure 1) The empirical results we describe below confirm that the increasing $\Delta E$ values did result in increasing perceptibility of the emphasized data element (e.g., see Fig 4).

For size, we chose eight fixed size differences (geometric difference in mark area between emphasized and non-emphasized content) from 25% to 200%. As shown in Figure 1, the size differences indicate area rather than diameter. Since previous research has also determined that perceived size can be different from geometric size, we then calculated the perceptual size difference for each level, and used the perceptual value in our analysis (again, we note that we do not currently have a correspondence between magnitude levels for

different variables, and so the levels that we chose simply provide a range within which we can gather empirical evidence about perception). We used the equation $Perceived\ size = Actual\ size^{0.86}$, were actual size signifies the area of the object, as suggested by previous work [38] (see Fig 1).

For blur/focus, we do not have a perceptual difference metric similar to colour's $\Delta E$ or to perceptual size as described above. Therefore, for blur/focus, we simply chose levels that cover a wide range of perceived prominence for all targets. We chose eight different blur intensities (applied to the non-emphasized areas of a visualization) implemented using GIMP's Gaussian Blur function – with blur radius ranging from 1 to 8. Examples of emphasized targets and corresponding distractors are shown in Figure 1.

Our first study measured the baseline perceptibility of the three emphasis effects at each effect's eight different magnitude levels, using artificial static scatterplot visualizations rendered using Chart.js[1]. Scatterplots were rendered on a white background using one-pixel gray axes. The second study used the same three effects, but only three of the eight levels; we used the visual variable to manipulate elements in visualizations taken from the MASSVIS dataset.

### 3.2 Apparatus

To record eye movement and interaction data we used an SMI Red-m eye tracker running at 60 Hz on a Dell 24-inch monitor (screen resolution of 1920x1080) connected to a Windows 10 PC. The viewing distance was approximately 60 cm (Figure 2). Gaze data was recorded using SMI Experiment Center and analyzed with SMI BEGAZE software. Users' heads were not fixed, but they were instructed to avoid unnecessary head movements. The experiment was conducted in an indoor laboratory with normal lighting conditions. All questionnaire data was collected through web-based forms.

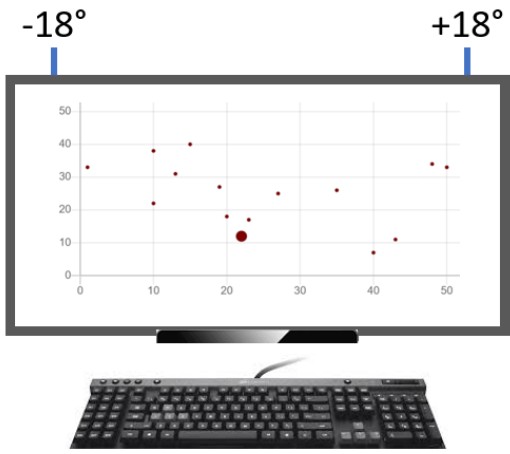

-18°  +18°

Figure 2: Setup and visualization graphic presented to participant for one trial.

## 4 STUDY 1: ESTABLISHING A BASELINE FOR PERCEIVED EMPHASIS

### 4.1 Participants

Twenty-one participants were recruited from the local university pool. We excluded three participants from our analysis either for self-reporting a colour vision deficiency, or for high eye-tracking deviation; this left eighteen people (7 male, 11 female) who were given a $10 honorarium for their participation. The average age of the participants was 26 (SD 4.5). All participants continuing to the study reported normal or corrected-to-normal vision and no colour-vision deficiencies, and all were experienced with mouse-and-windows applications (10 hrs/wk). Six participants reported previous experience with information visualizations from previous university courses.

### 4.2 Study Procedure

Participants completed informed consent forms and demographic questionnaires. Participants then completed a colour vision test: we checked for colour-vision deficiency using ten Ishihara test plates [31]. Next, we used the five-point calibration procedure from the SMI experimental suite to calibrate the eye tracker. Once the eyetracker calibration step was completed, participants carried out a series of trials with our scatterplot visualizations. The instructions given to participants were to visually explore each visualization and click on the element they felt was most emphasized. The monitor was blanked after each trial (after the participants clicked on an element) and the study software then asked the participant to rate the perceived visual prominence of the target mark, on a 1-7 scale. After participants provided their subjective rating, mouse position was re-centered and the next trial began.

To prevent learning effects and to account for attention theories that suggest that visual attention can be guided by previously seen targets [1], participants were assigned to a random order presentation of all visual stimuli. Each visualization contained one target mark (an emphasized stimuli) and twenty randomly-placed distractor marks, avoiding overlaps. While spatial distance between distractor marks and the target can influence colour difference perceptions [9], we elected to construct our scatterplots with variable element spacing to increase the visual complexity of the stimuli for increased ecological validity, while ensuring distractors and targets avoided overlaps. The three emphasis effect types were presented at their

$^1$https://www.chartjs.org/

8 magnitude-of-difference levels, and each emphasis level was presented 5 times. Each target maintained the same appearance for each of the 5 trials of the level, but changed location. Across all levels and locations, each visual variable was presented in 40 trials, all participants completed the 120 trials of the study. Our study setup ensured that targets were located at a maximum of 18° viewing angle from the centre of the screen (a previous study shows that participants are able to find targets at angles of 18° at least 75% of the time in under 240ms [21]). Our target positioning and the free-viewing method used in the study meant that participants were always able to keep targets close to their central vision. We changed the target's location to evaluate the visual variables at multiple locations within a visualization, while ensuring targets would remain visible in a participant's visual field.

### 4.3 Study Design and Analysis

Our goals for the study were to gather empirical data about perceptibility for the different visual variables and magnitude levels, and to use that data for a predictive model. To investigate differences, we used a repeated-measures design with two factors:

- Emphasis Effect (blur/focus, colour, size)

- Magnitude of Difference (levels 1-8)

Because magnitude levels are specific to each visual variable (e.g., the scale for colour difference is independent of the scale for size or blur), the levels are intended only to test increasing magnitudes for each effect. The ANOVA analysis for Magnitude therefore investigates whether perception of the visual variable is affected by each variable's increasing magnitude.

We used four dependent measures that provide different aspects of the user's experience with emphasis: first, we tracked time to eye fixation on the target (which provides an indication of early visual attention); second, we recorded the time to the user's mouse click on the target (which measures the user's conscious decision about emphasis); third, we recorded the user's total fixation time on the target (to determine whether stronger emphasis leads to longer fixation); and fourth, we asked for the user's subjective rating of the target's emphasis (which provides a more detailed measure of the user's conscious decision).

We used our analysis results to explore relative differences between the emphasis effects, and to fit curves to our empirical data in order to develop a predictive model.

## 5 STUDY 1 RESULTS

### 5.1 Time to Target Fixation, Time to Target Click, and Fixation Times

We analyzed differences between emphasis effect and magnitude of difference on participant's time to target fixation, target click, and fixation time in an Area of Interest (AOI) surrounding the emphasized visual target. We report effect sizes for significant RM-ANOVA results as general eta-squared $\eta^2$ (considering .01 small, .06 medium, and $>.14$ large [41]). For all follow up tests involving multiple comparisons, the Holm correction was used.

*Time to Target Fixation.* RM-ANOVA showed significant main effects of *Emphasis Effect* ($F_{2,34} = 17.73, p < 0.001, \eta^2 = 0.07$), and *Magnitude of Difference* ($F_{7,119} = 8.23, p < 0.001, \eta^2 = 0.19$) on time to target fixation, with no interaction between the factors ($F_{14,238} = 2.71, p = 0.27$). These data are shown in Fig 3; note that size levels for size are adjusted in the charts to show perceptual size (but for analysis, size levels were mapped to the standard 1-8 scale). Across all Magnitudes, participants fixated on targets fastest in the Blur/Focus condition (828 ms), followed by Size (913 ms) and Colour (1242 ms). Post-hoc t-tests showed significant ($p < 0.01$) differences between Color → Blur and Color → Size. Across all emphasis effects, time to target fixation was the fastest at magnitude

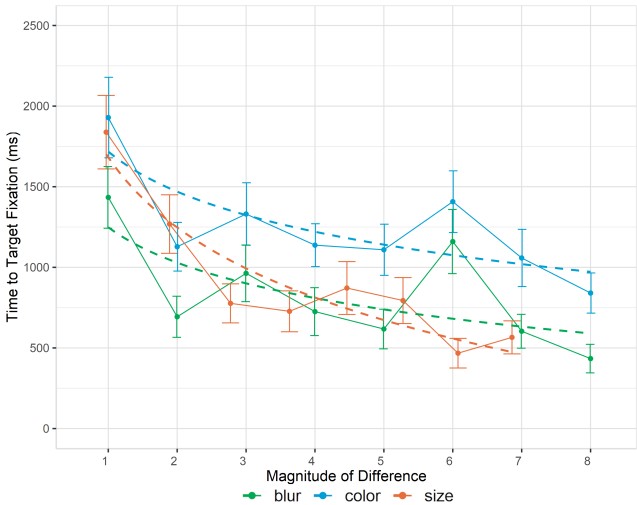

Figure 3: Time to Target Fixation. Empirical means (solid lines) and log curve (dashed lines). $R^2$ values for logarithmic curves: blur = 0.45, colour = 0.60, and size = 0.87. Levels for size are adjusted in the chart to show perceptual size rather than geometric size.

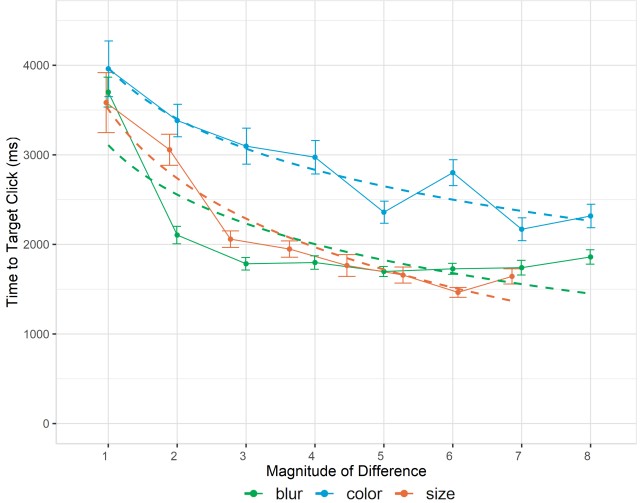

Figure 4: Time to Target Click. Empirical means (solid lines) and log curve (dashed lines). $R^2$ values for logarithmic curves: blur = 0.68, colour = 0.90, and size = 0.96. Levels for size are adjusted in the chart to show perceptual size rather than geometric size.

8 (613 ms), and the slowest at magnitude 1 (1733 ms). A similar post-hoc t-test was applied for pairs of magnitude differences and showed a significant difference for $1 \rightarrow$ 2-8 (p<0.001), and $3 \rightarrow 7$ (p<0.05).

*Time to Target Click.* RM-ANOVA showed significant main effects of *Emphasis Effect* ($F_{2,34} = 40.99, p < 0.001, \eta^2 = 0.24$), and *Magnitude of Difference* ($F_{7,119} = 56.45, p < 0.001, \eta^2 = 0.45$) on target click and an interaction between the factors ($F_{14,238} = 2.68, p < 0.01, \eta^2 = 0.08$). These data are illustrated in Fig 4. Participants clicked on focused targets fastest in the Blur condition (2051 ms), followed by size (2141 ms) and colour (2882 ms). Post-hoc t-tests again showed significant ($p < 0.01$) differences between Color $\rightarrow$ Blur and Color $\rightarrow$ Size. Averaged across all emphasis effects, time to click was fastest at magnitude 7 (1791 ms), and the slowest at magnitude 1 (3748 ms).

*Total Target Fixation Time.* RM-ANOVA showed a significant main effect of *Magnitude of Difference* ($F_{7,119} = 6.65, p < 0.001, \eta^2 = 0.12$) on fixation time, but no difference between *Emphasis Effects* ($F_{2,34} = 2.76, p = 0.08$). Averaged across magnitude, total fixation time was similar among the emphasis effects (1990ms for size; 2260 ms for both size and colour). Averaged across all effects, a Magnitude of 7 had fixation time of 2470ms, while a Magnitude of 1 had the least time at 1913 ms. Post-hoc t-tests showed significant (all $p < 0.05$) differences on difference pairs $1 \rightarrow 5$, $1 \rightarrow 7$, $3 \rightarrow 7$, and $7 \rightarrow 8$.

## 5.2 Subjective Perception of Visual Prominence

After the presentation of each visualization, participants were asked to rate how visually prominent the emphasized data point appeared to them. Mean response scores are shown in Figure 5. We used the Aligned Rank Transform [68] with the ARTool package in R to enable analysis of the subjective prominence responses using RM-ANOVA. For subjective ratings of perceived emphasis there were main effects of *Emphasis Effect* ($F_{2,408} = 56.38, p < 0.001$) and *Magnitude of Difference* ($F_{7,408} = 24.98, p < 0.001$), with no interaction ($F_{14,408} = 1.43, p = 0.13$). Results from these analyses follows those from Time to Click, in which sharp objects in the Focus/Blur emphasis condition were, on average, perceived as most visually prominent, followed by Size and Colour - with an increasing perceived visual prominence as we increase the Magnitude of

Difference.

## 5.3 Re-assessing Perceptual Levels for Color and Size

The magnitude levels chosen for color used the $\Delta E$ scale that is intended to provide perceptually-equal magnitude differences; similarly, the (adjusted) levels for size also provide linear magnitude changes in perceptual space. This means that we would expect a linear change in perceptual response to the changes in color and size (in Figures 3, 4, and 5). However, these charts show curves for both color and size, not linear relationships. This suggests that for visualization tasks, the perceptual scales for color and size may underestimate the decrease in emphasis as the effect increases in magnitude.

## 5.4 Participant Preferences and Comments

At the end of the study session we asked participants to state which emphasis effect they felt was the most visually prominent, least prominent, and to provide further comments on their responses. Overall, focus/blur was seen as most prominent (seven participants voted for blur/focus, six for size, and four for colour). One participant stated that no one effect stood out as most prominent. Participant comments for the three emphasis effects reflect the empirical findings, favouring blur/focus. One participant reported, "[In focus/blur] other data points were very blurry and hard to distinguish so the clear one stood out more than if the colour were different or the size were different (i.e. could only focus on the emphasized one, compared to the other types where you could still view the non-emphasized points)". Another participant stated "[blur/focus] clearly hid the other circles". Participants that favoured size reported that size may be easier for quick comparisons; one remarked "It is easier for the eye to visualize a bigger/smaller size in comparison to other dots vs trying to see a colour difference of a similar size dot".

## 5.5 Building an Initial Equivalence Model

We used the raw data from Study 1 to build initial predictive models of time to target fixation, time to click, and subjective rating of emphasis – and although more data will be needed to refine the predictions, we are able to capture some of the main differences between the three emphasis effects that we examined. Our models are simple functions fit to the raw empirical data; we use logarithmic functions they are commonly used to describe human performance

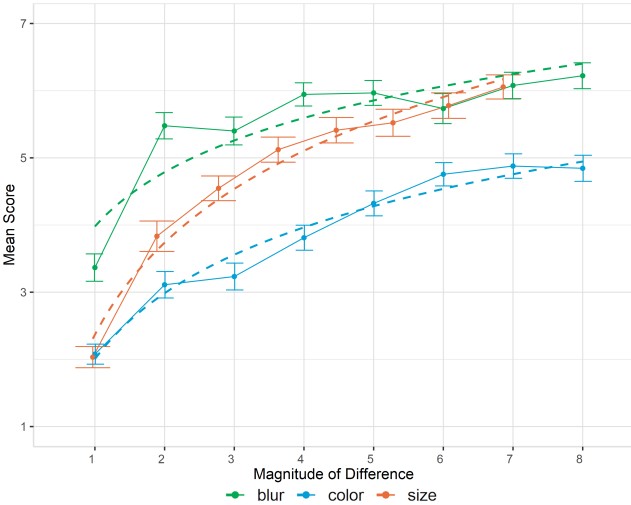

Figure 5: Perceived Prominence of Emphasis Effects. Empirical means (solid lines) and log curve (dashed lines). $R^2$ values for logarithmic curves: blur = 0.80, colour =0.97, and size = 0.98.

in signal-detection and perceptual studies [67]. We fit the functions to the data using R (lm(mean $\sim$ log(magnitude of difference)); we could then use R's 'predict' function to get predicted values. The fitted logarithmic curves for time to target fixation, time to click, and subjective ratings are shown in Figures 3, 4, and 5. Captions for these figures also state the $R^2$ values for the accuracy of the fitted functions to the data: for time to fixation the curve was only moderately accurate, but for time to click and subjective ratings, the accuracy was much higher.

The logarithmic curves provide a simple model that allows investigation of equivalence between the three effects. For all three measures, the models allow us to observe some main features of the relationships: first, colour is consistently less perceptible than the other two effects, both in terms of performance data and subjective ratings; second, size and blur/focus are very similar at level 3 and above of both performance measures, but at levels 1 and 2, size is somewhat weaker; third, size and blur/focus are more clearly separated in subjective ratings, with clear differences up to level 5.

These models, once validated, can allow simple calculation of equivalence between effects. As an example of how the calculation works, consider a scenario where a designer needs to change from a blur/focus emphasis effect to one that uses colour; interpolation of the curves of Figure 4 indicate that to translate the perceived emphasis of level 1 of blur/focus, a designer would need to use a colour effect of approximately level 7. However, before we can consider using the models for equivalence, we need to verify that they are robust enough to work with other visualizations. We do this by predicting data from Study 2 with the models developed from Study 1, as described below.

## 6 STUDY 2: PERCEPTION OF EMPHASIS IN COMPLEX VISUALIZATIONS

In contrast to the scatterplots used in Study 1, many visualizations include other visual factors such as background graphics, labels, titles, annotations and other embellishments that may affect how a user's attention is guided and ultimately how an emphasis effect is perceived. Therefore, we need to understand how users perceive emphasis effects in more complex visualizations. We designed our study following a similar method to Study 1, but evaluated emphasis effects in complex, real-world visualization graphics from the MASSVIS database [7].

### 6.1 Image Data

As the emphasis effects we are studying are not particularly targeted towards a specific visualization type, we chose the MASSVIS database [7] as the source for image data. The dataset contains 5000 static data visualizations that are obtained from a variety of online sources, are generated from real-world applications, and are targeted to a broad audience; MASSVIS is a popular choice for investigating how general users understand data visualizations. We selected a subset of 16 visualizations from the dataset covering a variety of visualization types, including maps and scatterplots. All of the selected graphics were chosen based on having scatterplot-like features (e.g., points on a map) to ensure consistency across our two studies. Each of the 16 visualizations had one emphasis effect applied at a time (Fig 6), and these augmented images were used to evaluate how users perceive the different emphasis effects.

### 6.2 Magnitude Levels for Emphasized Stimuli

We used a subset of Study 1's magnitude levels for Study 2 – we chose three uniform steps (1, 4, and 7) from Study 1, giving us coverage of the range we used in the baseline results. Example graphics with an emphasized data point are illustrated in Fig 6.

### 6.3 Experimental Design and Procedure

The experiment followed a similar procedure to that of Study 1. After providing informed consent and going through the eye-tracker calibration, participants were instructed to explore each visualization and to click on the area they felt was most emphasized. Similar to Study 1, to prevent learning effects and pre-attentive processing of previously seen stimuli, participants were assigned to a random order presentation. Test graphics contained one randomly-placed test mark in the graphic. After each stimulus presentation, participants were asked to rate the perceived visual prominence of the emphasized point they selected. Given the variety of colours and mark sizes in our sampled visualizations, our test mark colour and size difference are relative to those of the marks in each visualization. Individual differences for each image are considered in our discussion section. Each visual variable was presented in 48 trials (3 levels X 16 graphics), and all participants completed all 144 trials of the study.

### 6.4 Participants

Twenty-four new participants (none of whom participated in Study 1) were recruited from the local university pool. We excluded four participants from our analysis for high eye-tracking deviation, or failure to follow experiment instructions. The remaining twenty participants (9 male, 9 female, 2 non-binary) were given a $10 honorarium for their participation. The average age of the participants was 26 (SD 6.02) and all reported normal or corrected-to-normal vision and no colour-vision deficiencies; all were experienced with mouse-and-windows applications (10 hrs/wk), and 6 had previous visualization experience. We used the same experimental setup described in Study 1.

### 6.5 Study Design and Analysis

To understand how users perceive emphasis effects in more complex, real-world visualizations, and to asses our initial equivalence model, we used a repeated measures, within participants design with the same two factors as Study 1:

- Emphasis Effect (blur/focus, colour, size).

- Magnitude of Difference (Levels 1, 4, and 7 from Study 1).

We used the same four dependent variables: time to fixate on target, time to target click, total fixation time, and subjective rating of perceived emphasis.

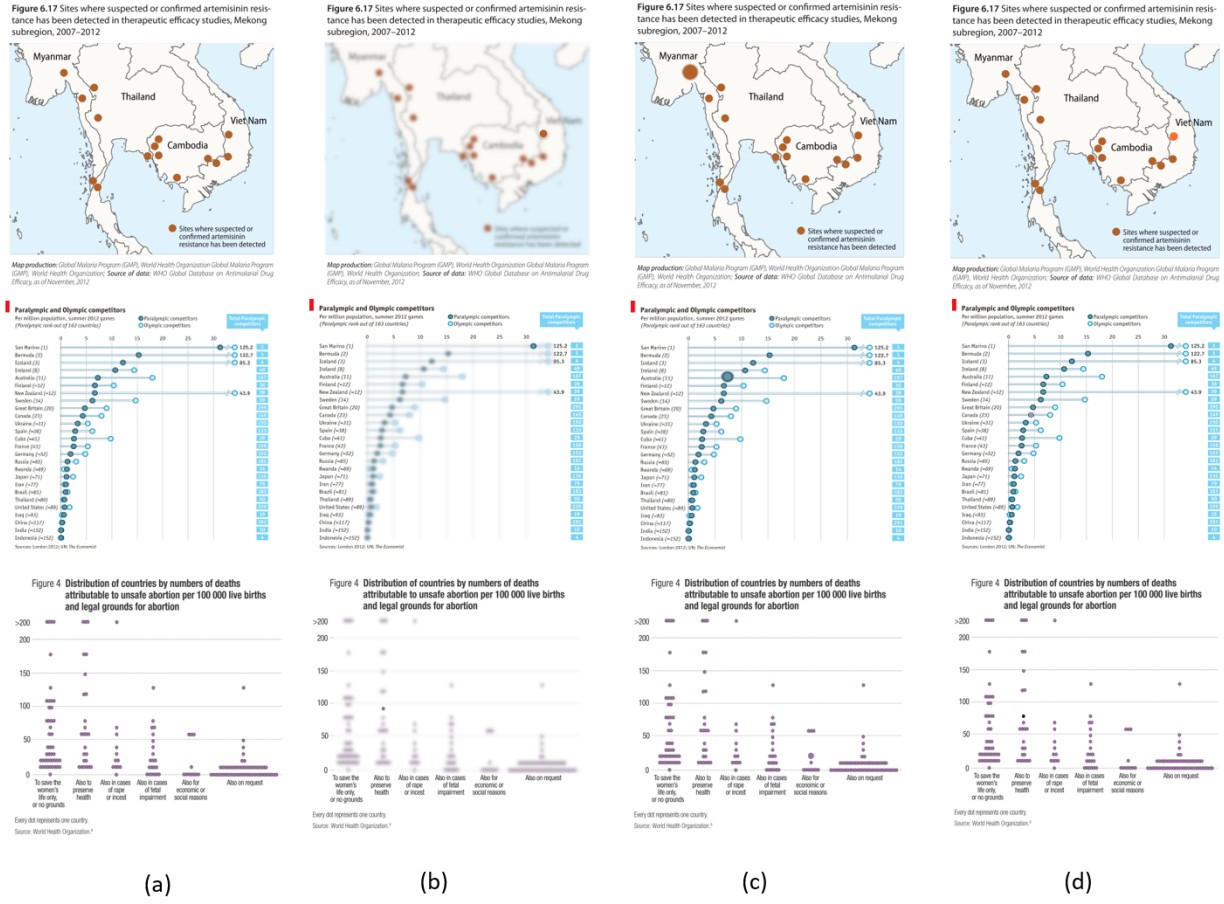

Figure 6: Example stimulus display for study 2 (a) baseline, (b) focus/blur, (c) size/area, (d) colour; Rows 1-2 at level magnitude of difference 4. Row 3 shows magnitude of difference 7.

## 7 STUDY 2 RESULTS

We again analyzed emphasis effect and magnitude of difference on our four dependent measures, and we again report effect sizes as general eta-squared $\eta^2$, and use Holm correction for followup tests.

*Time to Target Fixation.* RM-ANOVA found no main effect of *Emphasis Effect* ($F_{2,38} = 1.78, p = 0.18$) on time to target fixation, but did find an effect of *Magnitude of Difference* ($F_{2,38} = 3.80, p < 0.01, \eta^2 = 0.31$), and an interaction between the factors ($F_{4,76} = 3.09, p < 0.01, \eta^2 = 0.05$). These data are shown in Fig 7. Post-hoc t-tests showed significant ($p < 0.01$) differences between each magnitude-of-difference pair. Averaged across all emphasis effects, time to click on an emphasized data point was fastest at Magnitude 7 (3548 ms), and the slowest at Magnitude 1 (4932 ms).

*Time to Target Click.* RM-ANOVA showed significant main effects of *Emphasis Effect* ($F_{2,38} = 41.18, p < 0.01, \eta^2 = 0.32$) and *Magnitude of Difference* ($F_{2,38} = 58.04, p < 0.01, \eta^2 = 0.62$) on target click time, and an *Emphasis Effect* × *Magnitude of Difference* interaction ($F_{4,76} = 14.64, p < 0.01, \eta^2 = 0.15$). These data are illustrated in Fig 8. Similar to Study 1, focused targets in the Blur condition were clicked on fastest (3106 ms), followed by Size (4812 ms) and Colour (5787 ms). Holm-corrected post-hoc t-tests showed significant ($p < 0.01$) differences between all pairs. Averaged across all emphasis effects, time to click on an emphasized data point was fastest at magnitude 7 (4616 ms), and the slowest at magnitude 1 (7012 ms).

*Total Target Fixation Time.* RM-ANOVA showed a significant main effect of *Emphasis Effect* ($F_{2,38} = 3.41, p < 0.01, \eta^2 = 0.04$)

and *Magnitude of Difference* ($F_{2,38} = 15.08, p < 0.01, \eta^2 = 0.22$) on total fixation time, and a *Emphasis Effect* × *Magnitude of Difference* ($F_{4,76} = 4.30, p < 0.01, \eta^2 = 0.08$) interaction. Averaged across magnitude of differences, fixation time for blur/focus was 1369ms and 1240 ms for both Size and Colour. Averaged across all effects, a magnitude of difference of 7 gathered the most attention with a fixation time of 1494ms, while a difference of 1 had the least fixation time at 1080 ms. Post-hoc t-tests showed significant (all $p < 0.01$) differences for *Magnitude of Difference* but no difference among *Emphasis Effects*.

### 7.1 Subjective Perception of Visual Prominence in Complex Visualizations

After the presentation of each visualization, participants were asked to rate the visual prominence of the emphasized data point. Mean response scores are shown in Fig 9. We used the Aligned Rank Transform [68] with the ARTool package in R to enable analysis of the subjective responses using RM-ANOVA. RM-ANOVA showed there were main effects of *Emphasis Effect* ($F_{2,171} = 16.05, p < 0.001$) and *Magnitude of Difference* ($F_{2,171} = 60.00, p < 0.001$), and an interaction between the factors ($F_{4,171} = 2.57, p = 0.03$). Results from these analyses are shown in Figure 9 and follow those from Time to Click, in which sharp objects in the Focus/Blur effect were perceived as most visually prominent, followed by Size and Colour - with an increasing perceived visual prominence as the magnitude of difference increased.

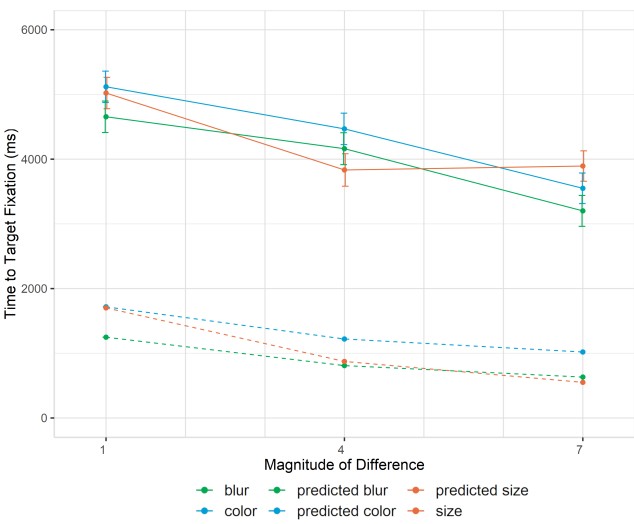

Figure 7: Time to Target Fixation in Complex Visualizations. Empirical means (solid lines) and predicted means (dashed lines).

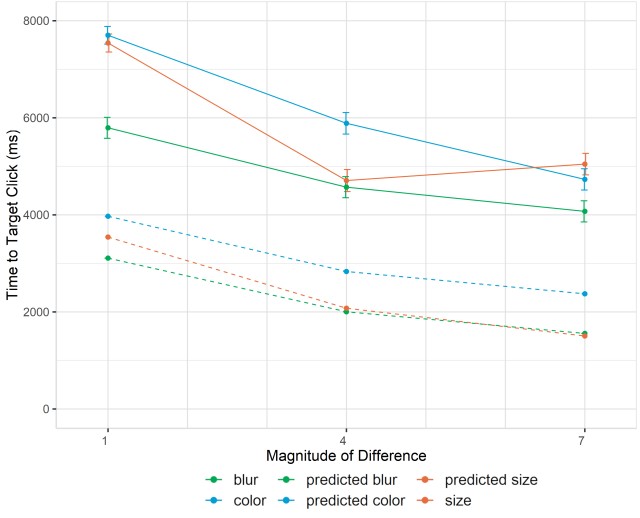

Figure 8: Time to Target Click in Complex Visualizations. Empirical means (solid lines) and predicted means (dashed lines).

## 7.2 Participant Preferences and Comments

After completing the study, participants provided their preferences and general comments on the emphasis effects they identified. Participant comments echoed our other findings. Participants made several comments on how the focus/blur emphasis effect helped them to rapidly identify content: for example"It [the emphasized point] just popped out more than the rest, provided more contrast"; "Because it [blur/focus] didn't allow me to see the others, I focused all my attention to the point that was not blurry". One participant favoured size, stating "[size] always drew my eye immediately".

When asked whether there were other areas of a visualization that got their attention, one participant remarked "The titles and information, I was trying to read them and see if that would have helped somehow to identify what was emphasized", while another participant stated "I occasionally looked at the titles to see what the information was representing".

## 7.3 Consistency Across Studies and Model Validation

We used the models built from Study 1 data to predict the data gathered for each effect and magnitude used in Study 2, and then compared the empirical data points to the predicted values (predictions are shown in Figures 7, 8 and 9 as dotted lines). Although the absolute values of the predictions are lower than the true values, the predictions do capture many of the characteristics of the Study 2 results, as discussed below. We tested the correlation between the predicted and empirical values: for time to target fixation, the correlation was 0.82 ($R^2$=0.87); for time to target click, correlation was 0.92 ($R^2 = 0.94$); for subjective ratings, correlation was 0.96 ($R^2 = 0.96$).

If equivalence models are to be useful, the perceptibility of emphasis must be reasonably reliable across different visualization situations. Our two studies involve two visual settings: plain scatterplots in Study 1, and more complex visualizations in Study 2 (with background graphics and colours, text, and multiple visual styles). Nevertheless, there are several similarities between the two sets of results (as indicated by the very strong correlation scores). In both studies, the colour effect was less perceivable (higher time to target fixation and target click time, and lower subjective ratings); however, the earlier difference between colour and size at the highest magnitude is now gone. As in study 1, the blur/focus effect is again consistently more perceivable (and is rated as more prominent). Also as in Study 1, there was a similar improvement in performance as the magnitude of the effect increases; there was less of a clear

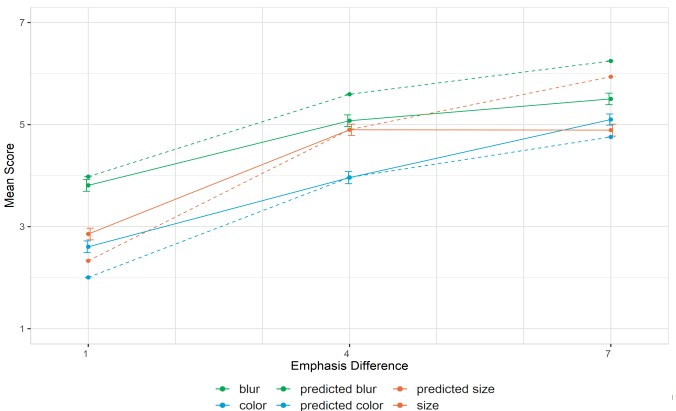

Figure 9: Perceived Prominence of Emphasis Effects in Complex Visualizations. Empirical means (solid lines) and predicted means (dashed lines).

logarithmic curve for the emphasis effects (although this would be less apparent with only three magnitude levels in Study 2).

The most obvious difference between the predicted and real values is that times for both fixation and clicking were substantially higher with the MASSVIS visualizations. However, this was an expected difference because of the additional visual information available in each image – and because all emphasis effects were affected similarly, any equivalence calculations using the model will be unaffected.

The subjective responses were particularly well predicted by the Study 1 model (see Figure 5), with the predicted points being accurate both in terms of absolute score and the relationship between the effects. This is a particularly valuable finding, because as discussed below, it may be that the user's perception of emphasis is a more important measure for designers than the user's gaze patterns or click behaviour.

The main point where the predictions were inaccurate – both for performance data and for subjective ratings – was the perceptibility of the size effect at level 7. After reviewing our stimuli for this condition, there are two possible reasons for the empirical results being different from predicted values. First, two of the visualizations (see

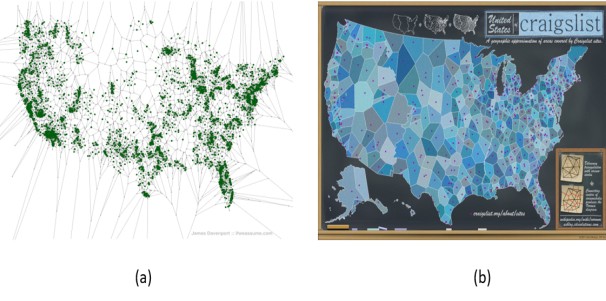

|     (a)     |     (b)     |

Figure 10: Study 2 graphics. Graphic (a) contained a larger number of data points with composite blobs, leading to visual crowding. Graphic (b) has multiple visual elements (shapes, colours, and text), reducing the effect of size emphasis on a data point.

Figure 10) contained a large number of data points and many visual elements overall, and previous research has shown that it is more difficult to recognize objects in a cluttered environment due to *visual crowding*, which can create a visual-perception bottleneck [40]. Second, when data points in these visualizations are dense, composite blobs with several overlapping points create marks that are larger than the default size. Although none of our target elements were in or beside these blobs, the presence of varying-size elements in the visualization may have reduced the prominence of our manipulation and forced participants to do a more careful visual search.

This anomaly with the size effect points to another useful aspect of having a predictive model, however: that is, the identification of empirical results that are not as expected and that may need to be investigated further.

## 8 DISCUSSION

### 8.1 Summary of Findings

We investigated how users perceive colour, size, and blur/focus when used as emphasis effects in both basic scatterplots and more complex visualizations. Magnitude scales for each variable are different, since we do not have a way to translate perceptual equivalency across effects, investigating this equivalency is one of the goals of our studies. Our evaluations provide several findings:

- *Different effects are perceived differently.* Across both studies, blur/focus led to fastest target fixation and target click, and was rated highest in terms of visual prominence by participants; size also led to fast performance and high ratings of visual prominence at higher magnitude levels (with one exception); colour led to the slowest performance and lowest ratings for prominence.

- *Magnitude increases emphasis, although with diminishing effects.* Across both studies, increasing the magnitude of the effect consistently increased visual prominence (again, with the same one exception); all effects showed a tailing-off of the effect of magnitude.

- *Models of perceived emphasis work reasonably well.* A predictive model based on logarithmic curves fit the Study 1 data well, and was reasonably accurate at predicting emphasis in Study 2 (particularly the subjective ratings).

In the following sections, we consider possible explanations for these results, look at how our findings and models can be used to assist designers in building visualizations with emphasis, and discuss limitations and directions for future research in this area.

### 8.2 Explanation of Results

*Differences Between the Emphasis Effects*

We saw consistent differences in fixation time, click time, and subjective ratings for our three emphasis effects, and the reasons for these differences arise from each technique's fundamental properties (as introduced earlier). First, blur/focus is an effect that manipulates the entire visualization except for the emphasized data element, and so has advantages over single-element techniques like colour and size. In particular, the blur effect guarantees that there will be no inadvertent competing visual stimuli that could slow the user's visual search (as happened with size at level 6 in Study 2), because all other elements are blurred. Second, the relative advantage for size over colour in our studies can be explained by the inherent limit on colour difference (i.e., there is a maximum difference between any two colours) whereas size difference has an unlimited upper end. The study results show that our range of magnitudes for size was larger than our range for colours – which points to the need for a better understanding of equivalences between effects.

*Effectiveness of the Predictive Model*

The model built from Study 1 data provided accurate predictions of the results in Study 2 ($R^2$ values of 0.87, 0.94, and 0.96 for fixation time, click time, and subjective rating), and the model correctly represented the overall relationships between the emphasis effects and the changes expected with increasing magnitude level. The success of the predictive model shows that perception of emphasis is consistent between our two experimental settings – plain scatterplots in Study 1, and realistic scatterplots with other visual features in Study 2. In addition, the model was a useful tool for identifying results that need further exploration, including the greater overall response times for the MASSVIS dataset, and the anomalous performance of size at level 7. Of course, these anomalies can only be spotted when there are empirical results to compare to the model, but it is likely that the model will be used together with empirical testing until it matures with the addition of more data in different settings.

*The Size-at-Level-Seven Anomaly*

As described above, the size effect at level seven was less prominent than expected, with two possible reasons: visual crowding from other elements in the visualizations, and inadvertent size variance from overlapping data points (see Figure 10). This result clearly indicates that there can be emergent properties in real-world visualizations that interfere with the user's perception of emphasis, and thus a planned emphasis effect must be considered in light of other visual elements. These real-world interactions are another motivation to have equivalence metrics, so that designers can switch from one emphasis effect to another (and preserve the prominence of the emphasized element) when interference is discovered.

While our setup of the presentation of visual stimuli ensured that distractor marks and stimuli would not overlap, changes in the distance between distractors and the emphasized elements may affect their noticeability. Because effects of visual crowding occur with a wide range of objects, colours and shapes [66], this phenomena may have affected other individual data points as well; but our explicit decision to not control the distance between points in Study 1 means that our results provide a more valid representation of the challenges faced by designers when emphasizing elements in a crowded visualization. As noted above, global effects such as blur/focus are less affected by visual crowding, as blurring non-targets partially eliminates them from a user's view, leaving only the focused element available. We note that it is possible to quantify the overall degree of visual complexity in an image, and in future work this could be added to our models as a factor (i.e., further studies could examine perceived emphasis at different levels of crowding).

A final reason that needs to be considered is the existence of possible noise in our results. We can see similar patterns from Study 1 with certain levels of a variable performing similar or worse than the previous lowest level (such as Size level 5 being slower than level 4). However, these differences are small (300ms), and given our overall quick response times across our empirical measures, variations of approximately 500ms could be attributed to a normal variation in participant response times.

## 8.3 Implications for Design

Our findings are applicable in a number of different visualization contexts. Visualization designers often need to draw a user's attention to important data points; our studies improve understanding of how visual cues are detected as emphasis effects and offer insights to their perceived visual prominence. While the current set of visual stimuli examined was relatively small, we intend to explore further visual variables in future studies.

A first design implication is that global visual effects such as blur/focus can achieve a high perceived visual prominence and remain relatively unaffected by a visualization's background. Perceived differences for other variables such as colour and size can be affected by the non-target elements, but by blurring the non-target objects in a visualization, the focused item is less likely to be affected by visual crowding. In visualizations with a large number of objects (such as different colours and shapes), blurring non-targets may achieve the highest noticeability – however, blur/focus cannot be used in visualizations where the user needs to inspect elements that are not emphasized.

Second, predictive models of perceived visual prominence can be valuable tools for designers. Although our model is still only a first step, it was already able to predict the results of Study 2 reasonably well, and can already be used to consider the equivalence between perception of the three effects that we tested. (We note that the model should not be used to calculate exact conversion factors between the effects, but rather to understand general relationships and approximate relative magnitudes). As further studies are carried out and more data is added, models like ours can become resources for designers that can accelerate the design of a narrative visualization. It is interesting that the model was most accurate at predicting people's subjective ratings of prominence, which raises the question of which metric is most important. It may be that subjective perception is a better measure for a model, because when a designer adds emphasis to a visualization, they typically want the viewer to know that the item is being emphasized – that is, what the viewer thinks is being emphasized is possibly more important than what their eye is drawn to first.

A third design consideration is for designers utilizing colour as a way to emphasize certain data points. It should be noted that a subset of users suffer from various genetic conditions which cause atypical forms of colour perception - in such cases, a different emphasis effect may be more appropriate. Designers may wish to use our metrics and results to evaluate the effectiveness of a different visual effect to achieve the same perceived importance. Our future work intends to evaluate the use of various visual cues for emphasis effects and compare the sets for individuals with normal vision and users with a vision deficiency.

Beyond visualization, our findings can also be applicable in other domains. For example, interface designers may wish to use our results as a way of devising methods of providing visual feedback. For instance, visual feedback during "find" tasks in different software software such as web browsers and pdf readers varies - with some software opting for colour highlighting an item when found, while others increase its size or use a combination of both. To effectively guide a user's attention to an item, designers can use perceived visual prominence as a method to evaluate and compare different visual effects.

## 8.4 Generalizability, Limitations, and Future Work

Our studies tested a limited range of visualizations (i.e., scatterplot presentations), so the application of our results should be limited to that type; in Study 2, however, we did test a wide variety of different visual styles taken from real-world examples, and so we believe that our findings will be robust across a range of real scatterplots. In future work, we plan to extend our work to other types of visualizations and other real-world scenarios, with a variety of datasets. We also tested only a single emphasized data point, and an opportunity to extend to our work is to investigate visualizations that emphasize multiple points. Multiple points of emphasis also provides us with another opportunity to test the predictions of the model – that is, if two data elements are emphasized with different effects that our model predicts should be equally prominent, which will the user fixate on first? (We note that this kind of comparison is only possible with single-element effects such as size and colour).

The difference levels for the visual variables tested in our experiments are intended to be generalizable for the design of emphasized elements in typical visualizations. However, although we tested a wide range of magnitude of differences, it is possible that our findings are influenced by the magnitude of differences we tested (as noted above in terms of the range of difference that is possible with each visual variable). A variety of other visual variables can be implemented as emphasis effects (see Healey for a review [24]). Visualization designers that intend to use a different range of magnitude of differences or emphasis effects may follow methods similar to the ones presented in this paper - in particular, testing user's reaction times and their subjective ratings to determine the noticeability of their effects. We also plan to carry out studies that look at how magnitude of emphasis is affected by clutter and by other mappings of visual variables to data variables.

Other factors in generalization should be considered as well. Colour perception models rely on a simplified model of the world that assume perfect viewing conditions. While this assumption is necessary for understanding the visual system, complexities of the real world such as the viewing environment [37], lighting conditions [8, 48], and display device [53] may affect visual perception. Our experimental viewing conditions were controlled and remained stable throughout the studies, however, future work could extend these results to larger user samples and different viewing conditions, using crowd-sourcing methods [25].

There are several additional opportunities for extending our findings. We explored emphasis effects with static visual variables (time-invariant in terms of Hall et al.'s framework [22]) but there are many other effects that could be tested, including depth, outline, transparency, or shape. Additionally, future research should investigate time-variant emphasis effects with dynamic visual variables such as flicker or motion and extend our results to interactive visualizations.

We evaluated our emphasis effects based on empirical metrics such as time to target fixation, and time to mouse click. There are other ways emphasis effects can be evaluated. For instance, the MASSVIS dataset contains a comprehensive set of user attention maps on the visualizations [7]. We intend to analyze viewer's attention maps on the visualizations, comparing the visualization's attention maps with and without an emphasis effect applied.

Finally, we elected to use the CIE2000 as it is commonly used in visualization and has been methodologically validated in past studies [28, 61]. Future work may consider the use of other colour difference models or colour spaces, such as CIECAM02 [45]. We anticipate investigating a number of different colour spaces will result in more accurate models of colour difference perceptions for visualization design.

# 9 CONCLUSION

Emphasis is an essential component of InfoVis, and is used by designers to draw a user's attention or to indicate importance. However, it is difficult for designers to know how different emphasis effects will compare and what level of one effect is equivalent to what level of another when designing visualizations. We carried out two user studies to evaluate the visual prominence of three emphasis effects (blur/focus, colour, and size) at various strength levels, and developed a predictive model that can indicate equivalence between effects. Results from our two studies provide the beginnings of an empirical foundation for understanding how visual effects operate and are experienced by viewers when used for emphasis in visualizations, and provide new information for designers who want to control how emphasis effects will be perceived by users.

# 10 ACKNOWLEDGMENTS

This work was supported in part by the Natural Sciences and Engineering Research Council of Canada (NSERC), and additionally by a Canada First Research Excellence Fund (CFREF) grant sponsored by the the Global Institute for Food Security(GIFS).

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
