# OpenReview forum: "A Baseline Study of Emphasis Effects in Information Visualization"
_graphicsinterface.org/Graphics_Interface/2020/Conference — GI 2020_

### Official Review · AnonReviewer2 · 2020-02-12
**Improved version of the paper but concerns remain.**

**Rating:** 5
**Confidence:** 5

**Review:**

In their revisions the authors have addressed a number of the problems identified by the reviewers, but they have neglected to attend to several important flaws that should be corrected. These remaining issues continue to weaken the paper. The authors do not really tackle the difficult issue of distance in the users' reaction metrics to the different effects and magnitudes. There are both perceptual (as in foveal acuity) and interaction influences. At the very least, they could have considered standard Fitts' law models of reaction time to click the target.  There should be a more substantive discussion of the weaknesses inherent in this lack.  (By the way, i am not sure where the authors base their comment on "area is more perceptually noticeable than size". Noticeable aside, we are much poorer at making area quantity judgments as opposed to length judgments.)

Similarly, the authors claim they clarified how visualizations were chosen. They used 16 visualizations. Which ones??? I'd like to know exactly which forms were used. Saying they are :"like" maps and scatterplots is insufficient.    This complicates the still unsatisfactory experimental design description of Study 2 in the paper.   For example, they claim this is a repeated measures study with 3 effects x 3 levels. This suggests 9 distinct conditions. BUt in fact there are 16 visualizations, so now there are 9x16 = 144 distinct conditions.  They say they used a repeated measures design, indicating participants did MORE than one repetition of each condition. How many?  Did all participants carry out 144 trials in complete random order?
 Unclear.  a randomly generated order will handle only first-order effects and not second-order effects (where factors repeat.)   I don't want to be too picky, but if the authors made these compromises in design, they should DISCUSS how these may be limitations of the study design. It's almost impossible to claim robust results from this kind of study with these kinds of questions and conflating factors incompletely described.

---

### Official Review · AnonReviewer3 · 2020-02-16
**Some experimental aspects well addressed (but still minor issues), others need a little work.**

**Rating:** 7
**Confidence:** 5

**Review:**

The new revision of the paper has clarified several issues. For example the points that were well addressed are the addition of information on participants, the reporting of p-values, and for the most part the equivalence of perceptual levels (see minor 2-3 things to fix in the end).

I still believe this paper makes a good contribution, has interesting findings and is of interest to our community. I am overall in favour of acceptance at this point if the list of issues mentioned next are addressed for the camera ready:

1. The addition of past work was not well integrated. For example, the paper added a reference on past work by Haley & Enns [24] but did not explain how this influences/informs designs or explains results. Similarly, the addition of Duncan & Humphreys [18] has implications about the study for creating the model (choice of distractors), but are not really discussed. A deeper reflection as to the relationship of this work to these included papers work would  be good (although many more references were mentioned by reviewers and do not seem to be added).

2. I am overall happy with how the differences in perceptual magnitude were treated in the reporting of the paper. I would suggest the following minor additions:
- p2: make the following its own paragraph => “It is important to note that the magnitude scales …”
- p2: remove the sentence “(since area is more perceptually noticeable)” this is not true, differences in area are notoriously harder to see than in length (diameter).
- p11: (section 8.1) Please stress here again the goal and lack of perceptual magnitude as some readers may not read the details (but will read the summary).

3. It would be good to further add the number of trials seen by participants to make sure all factors are reported (for example was it 3x8 in study 1 and 16x3x3 in study 2 as the text implies).

4. Finally, the paper now reports that the choice of visualizations in study 2 was based on their resemblance of scatterplots.  This makes sense given the creation of the model, but it would be good to have the list of them (and levels) in supplementary material or an appendix. (More generally, for such a study that raises opportunities for future research replication is important, so I would strongly encourage the authors to share their experimental material, results, and analysis scripts).

It would also be good to have a comment about possible differences across visualizations in study 2. This is a point that likely requires additional analysis and space to report formally, so I am personally ok if it is not discussed in detail or provided in supplementary material. It is more of a wish and research curiosity (if the materials are shared this would help).

---

### Official Review · AnonReviewer1 · 2020-02-18
**Review: A Baseline Study of Emphasis Effects in Information Visualization**

**Rating:** 6
**Confidence:** 3

**Review:**

This article investigates emphasis effects in information visualisation. It focuses on three emphasis strategies : varying colour, size, and blur/focus. The authors conducted two studies, the first one seeks to establish the effect of varying levels of salience of one emphasised element among many. Based on the measured effects the authors built a model for predicting the relationship between the emphasis effects. A second study was conducted with more realistic visualisations and is meant to test the model in more complex conditions.

The authors found that blur was the most effective strategy to improve prominence followed by size variation and colour rating the worst. The level of the emphasis has an impact on prominence as well. The model built from study one managed to accurately predicting perceived emphasis in the 2nd study (particularly subjective ratings).


As a non specialist, I found the paper interesting, and the studies tackling a complex and relevant problem. The emphasis effect chosen appear to be justified for a first study. Not being familiar with the use of distractors, I would have expected slightly more control in their design, or a discussion of the distractor design and default selected. While I may have missed it, I could not find the scale and question used for the subjective rating, which makes it particularly challenging to interpret the results.
The results of study 1 are insightful. They provide a basis to build upon in future research, and they can already be used to provide design guidelines. I appreciated the reporting of effect size using general eta squared.

Study 2 was conducted with more realistic visualisations. It provides more nuanced results on the effect of emphasis strategies. It would have been really useful to have a description of all the visualisations selected to better interpret the results. I would have expected that some visualisations would have been more suited for one type of emphasis or another. I regret the (relative) lack of discussion of emphasis management in “real-life” visualisations, whether some emphasis strategies can be more easily integrated in existing visualisations, or whether some types of emphasis strategies are more suited for one visualisation idiom or another (e.g. for online articles or for visual analytics tools).

Overall I found the paper insightful and the implications for design derived from Study 1 actionable. The methods section of both studies could be reformatted to make the study design stand out more clearly. And the discussion could open up to broader reflections on emphasis strategies in visualisation design.

---

### Meta-Review · Area_Chair1 · 2020-02-21

**Recommendation:** Accept
**Confidence:** 2

**Metareview:**

The reviewers found that the article tackles of problem of relevance in formation visualisation, and that the studies bring relevant insights to the community. They nonetheless highlight of number of problems remaining in the article that should be addressed before being published.

1. Clarify the presentation of the study conditions (all reviewers).
2. Better integrate of the literature in relation to the problem tackled in the article, both in the related work section (R1) *and* in the discussion (R2).
3. Clarify the visualizations used in study 2 (all reviewers).
4. Expand on the limitations of the experiments and their analysis (R2)
5. Reflect and contextualise of the results may apply in the wild (R3)

---

### Decision · Program_Chairs · 2020-02-21

Accept